# PagMYB205 Negatively Affects Poplar Salt Tolerance through Reactive Oxygen Species Scavenging and Root Vitality Modulation

**DOI:** 10.3390/ijms242015437

**Published:** 2023-10-22

**Authors:** Lieding Zhou, Xuhui Huan, Kai Zhao, Xia Jin, Jia Hu, Shuhui Du, Youzhi Han, Shengji Wang

**Affiliations:** College of Forestry, Shanxi Agricultural University, Jinzhong 030801, China

**Keywords:** *Populus abla × Populus glandulosa*, MYB transcription factor, salt stress, antioxidant enzymes, root vitality

## Abstract

Salt stress is one of the major abiotic stresses that limits plant growth and development. The MYB transcription factor family plays essential roles in plant growth and development, as well as stress tolerance processes. In this study, the cDNA of the 84K poplar (*Populus abla* × *Populus glandulosa*) was used as a template to clone the full length of the *PagMYB205* gene fragment, and transgenic poplar lines with *PagMYB205* overexpression (OX) or inhibited expression (RNAi, RNA interference) were cultivated. The role of PagMYB205 in poplar growth and development and salt tolerance was detected using morphological and physiological methods. The full-length CDS sequence of *PagMYB205* was 906 bp, encoding 301 amino acids, and the upstream promoter sequence contained abiotic stress-related cis-acting elements. The results of subcellular localization and transactivation assays showed that the protein had no self-activating activity and was localized in the nucleus. Under salt stress, the rooting rate and root vitality of RNAi were higher than OX and wild type (WT). However, the malondialdehyde (MDA) content of the RNAi lines was significantly lower than that of the wild-type (WT) and OX lines, but the reactive oxygen species (ROS) scavenging ability, such as the peroxidase (POD), superoxide dismutase (SOD), and catalase (CAT) enzyme activities, was dramatically more powerful. Most significantly of all, the RNAi3 line with the lowest expression level of *PagMYB205* had the lowest MDA content, the best enzyme activity and root vitality, and the best salt stress tolerance compared to the other lines. The above results suggest that the transcription factor PagMYB205 could negatively regulate salt stress tolerance by regulating antioxidant enzyme activity and root vitality.

## 1. Introduction

Soil salinization is one of the main environmental stresses limiting plant growth and production [1]. Research has shown that 20% of the world’s arable land and nearly half of the irrigated land are affected by salt stress in varying degrees [2]. Plants are typically exposed to many extreme environmental events during their lifetimes and have developed an elaborate and complex network of self-regulation to adapt to severe abiotic stress [3].

Transcription factors (TFs) are often key regulators of plant responses to biotic and abiotic stresses [4]. The perception and transduction of early salt stress signals in plants are mainly controlled by transcription factors such as DREB, bZIP, NAC/NAM, ATAF1, MYB, and others [5]. The overexpression of *GmDREB1* improved salt tolerance and leaf protein in response to high salt concentration in transgenic wheat [6]. The wheat C-bZIP gene *TabZIP14-B* was involved in the regulation of salt tolerance and frost resistance in transgenic plants, and overexpressing lines showed dramatical salt tolerance ability compared to the wild type [7]. NAC (NAM, ATAF1/2, CUC2) transcription factors were plant-specific and had distinct functions in plant developmental and abiotic stress response processes [8]. A stress-responsive gene, *PeNAC1*, was successfully isolated from poplar and played an important role in salt stress endurance by regulating Na^+^/K^+^ homeostasis [9]. The expression of *ATAF1* of the *Arabidopsis* NAC gene was significantly induced by high salt levels and abscisic acid (ABA) [8]. The *RmMYB108* gene was induced by low temperature, salt, and drought stress [10].

The MYB transcription factor family is one of the largest families of transcription factors in plants and plays an important role in the regulation of plant growth, development, and metabolism [11]. MYB TFs have highly conserved DNA binding domain (DBD) repeat sequences (Rs) in the N-terminal. In order to distinguish them from c-Myb protein, these repeats were renamed R1, R2, and R3 [12]. The R2R3-MYB transcription factor is the most abundant MYB protein in plants; it has a transcriptional activation domain at the C-terminus and plays key roles in plant cell differentiation, hormone response, secondary metabolism, and resistance to biotic and abiotic stresses [11,13,14]. For example, *MYB7* played a role in regulating carotenoid and chlorophyll accumulation in tissues by transcriptionally activating metabolic pathway genes [15]. *MdMYB308L* positively regulated cold tolerance and anthocyanin accumulation in apples by interacting with *MdbHLH33* [16]. *RmMYB108* responded to low-temperature stress in *Rosa multiflora* and cold hardiness in *Arabidopsis thaliana* [10]. The overexpression of *MYB12* and *MYB75* in transgenic *Arabidopsis* significantly increased the accumulation of flavonoids with potent antioxidant activity, resulting in enhanced tolerance to drought and oxidative stress [17]. *MYB94* and *MYB96* together activated epidermal wax biosynthesis to promote drought tolerance in *Arabidopsis* [18]. *MYB60* was specifically expressed in guard cells and enhanced plant drought tolerance by reducing water dissipation. The expression of *MYB60* was induced by signals that promote stomatal opening, such as white and blue light, but was inhibited by darkness, desiccation, and ABA treatment [19]. *PYL8* directly interacted with transcription factors *MYB77*, *MYB44*, and *MYB73* to increase the binding ability of *MYB77* to the target motif MBSI in the promoters of multiple growth factor responsive genes and promoted lateral root growth [20]. *MYB49* positively regulated salt tolerance by modulating stratum corneum formation and antioxidant defenses [21]. Many MYB proteins involved in the ABA response are also involved in regulating salt tolerance by regulating genes in the ABA signaling pathway [12]. *MYB12* overexpression induced the expression of ABA and proline biosynthesis-related genes, as well as improved flavonoid biosynthesis and reactive oxygen species (ROS) scavenging to confer salt tolerance in plants [17]. *MYB42* positively regulated salt tolerance by regulating the expression of *SOS2* (salt overly sensitive 2) [22]. *MYB30* regulated salt tolerance in plants by regulating mitochondrial alternative oxidase AOX1a (alternative oxidase 1a) [23].

Poplar is an important raw material for biofuels and is widely used as green forest, timber forest, and protection forest for production, road greening, and desertification control [24]. Currently, genetically modified poplar is still in field trials in the United States and Europe due to the rigorous environmental risk assessments required for regulatory approval. China is the only country in the world that has approved the commercial cultivation of genetically modified poplar [25,26]. As a result of land clearing, unsustainable irrigation practices, and pressure to bring marginal land into production, salinity stress has become an important environmental factor limiting poplar growth [27,28]. Therefore, there is an urgent need to breed new poplar species with salt tolerance. This study provides the theoretical and material basis to further reveal the function of PagMYB205 (*Potri.019G081500.1*) in plant abiotic stress endurance and the breeding of salt-tolerant, superior tree species.

## 2. Results

### 2.1. Gene and Protein Structure of PagMYB205

The CDS sequence of *PagMYB205* was 906 bp and encoded 301 amino acids. The molecular weight of the protein was 34.16 kDa, the molecular formula was C_1473_H_2348_N_436_O_472_S_14_, and the number of atoms was 4743 (Appendix A); PagMYB205 protein was a hydrophilic unstable basic secretory protein consisting of 20 amino acids with no transmembrane region (Appendix A and Appendix A). Protein structure analysis showed that the irregular helix was the most abundant region in the secondary structure of PagMYB205, followed by the α-helix, which accounted for 51.50% and 31.56%, respectively (Appendix A). Analysis of the cis-acting elements of the upstream promoter of *PagMYB205* revealed that the promoter contains abiotic stress-related elements, such as ABA response element (ABRE) and MeJA hormone response element, as well as some light response elements and seed-specific regulation elements (Appendix A).

### 2.2. Homologous Sequence Alignment

Based on the sequence alignment, nine species with a high homology of PagMYB205 were identical, and all of them had two MYB binding domains (Figure 1A). A phylogenetic tree was constructed, and PagMYB205 was more closely related to *Ricinus communis*, *Manihot esculenta*, and *Hevea brasiliensis* (Figure 1B). These results indicated that PagMYB205 belonged to the R2R3-MYB subfamily.

### 2.3. Localization and Self-Activating Activity Analysis of PagMYB205

The green, fluorescent signal of 35S:PagMYB205-GFP was mainly observed in the nucleus of tobacco protoplasts, while 35S:GFP was in the nucleus, cytoplasm, and cell membrane (Figure 2A). For the recombinant vector pGBKT7-PagMYB205, positive and negative controls were grown normally on the SD/-Trp medium, while pGBKT7-PagMYB205 could not grow on the SD/-Trp/-His/-Ade/X-α-Gal medium (Figure 2B). These results indicated that PagMYB205 played a transcriptional regulation function in the nucleus and had no self-activating activity.

### 2.4. Identification of Transgenic Plants

The 35S:PagMYB205-GFP and CAM-RNAi-PagMYB205 recombinant plasmids were transferred into 84K poplars (*Populus abla × P. glandulosa*) by using the *Agrobacterium* rhizogenes-mediated leaf disk method to cultivate *PagMYB205* overexpression (OX) and inhibited-expression (RNAi) transgenic lines, respectively. Transgenic seedlings that could root normally in 1/2 Murashige and Skoog (MS) medium containing 50 mg/mL Kanamycin and 200 mg/mL Timentin were selected as positive lines (Appendix A). The expression of *PagMYB205* in the 13 transgenic lines was then detected with qRT-PCR. As shown in Appendix A, *PagMYB205* was significantly up-regulated in nine OX lines compared to the wild-type (WT) line and significantly down-regulated in four RNAi lines. Three RNAi (RNAi1, RNAi2, and RNAi3) and three OX (OX1, OX2, and OX3) lines with high, medium, and low expression levels were selected for further physiological experiments, respectively (Figure 3A and Appendix A).

### 2.5. PagMYB205 Decreases Poplar Root Vitality under Salt Stress

When treated with salt stress, the rooting rate of the RNAi was higher than that of the other lines (Figure 3B). Then, we compared the root vitality of different lines under non-salt and salt treatments. The results showed that the root vitality of all other lines was not significantly different from WT except OX3 under non-salt treatment conditions, while the root vitality of RNAi (especially RNAi3) was significantly higher than that of WT under salt treatment (Figure 3C). However, there was no significant difference between OX and WT.

The morphological indicators were also measured, and the height of the RNAi plants was higher than that of the WT plants in the non-salt treatments (Appendix A). When subjected to salt stress, the root length, root area, specific root length, specific root area, root tissue density, and absorbed root length of OX, WT, and RNAi were decreased, but root diameter and volume were increased (Appendix A). At the same time, the stomatal density of OX, WT, and RNAi was slightly decreased under salt stress (Figure 4D). However, there were no significant differences in the other morphological indicators and stomata structure of leaves between non-transgenic and transgenic plants (Figure 4, Appendix A and Appendix A). All of these above results indicated that PagMYB205 may negatively regulate the root vitality of poplar to adapt to salt stress, but not the leaves.

### 2.6. PagMYB205 Interferes with ROS Scavenging under Salt Stress

Since superoxide dismutase (SOD), peroxidase (POD), and catalase (CAT) play an important role in ROS scavenging, we further determined the activities of SOD, POD, and CAT in each poplar line under both treatments. After salt treatment, the SOD, POD, and CAT activities of the RNAi were significantly higher than those of the other two lines, with RNAi3 having the highest enzyme activity (Figure 5A–C). The SOD, POD, and CAT activities of the OX were lower or equal to that of WT (Figure 5A). The malondialdehyde (MDA) content was measured to assess the extent of membrane lipid peroxidation under salt stress, and the results showed that the MDA content of OX was lower than that of WT under non-salt treatment but increased to the level of WT when treated with salt stress (Figure 5D) However, the MDA content of RNAi was less than WT under both non-salt and salt stress conditions. These results suggest that PagMYB205 negatively regulates the activity of plant antioxidant enzymes under salt stress conditions to interfere with the scavenging of ROS in plants.

## 3. Discussion

Plant growth and development are severely affected by abiotic and biotic stresses from the environment [29]. The secondary salinization of soil disrupts the dynamic balance of ROS in plants, causing a series of problems such as nutrient deficiency, oxidative stress, and ion imbalance, and has become one of the important environmental stressors that limits plant growth and development [30]. The transcription factor tightly regulates gene expression related to hormones, symbiotic interactions, cell differentiation, and stress signaling pathways of plants [31]. The MYB transcription factor family has been reported to be involved in a variety of processes, including primary and secondary metabolism, plant development, and responses to biotic and abiotic stresses [12,13].

In this study, abiotic stress-related cis-acting elements were detected in the promoter region of *PagMYB205* through bioinformatics analysis. The PagMYB205 protein was found to be an unstable hydrophilic soluble protein localized in the nucleus, without transmembrane structural domains and self-activation. The homology sequence alignment and phylogenetic tree indicated that *PagMYB205* belonged to the R2R3MYB subfamily that is involved in plant growth and development, primary and secondary plant metabolism, hormone regulation, and stress response stages [32]. In *Arabidopsis*, the R2R3MYB member *AtMYB44* responds to drought stress by participating in stomatal regulation and ROS accumulation [33,34]. *GmMYB118* maintains cellular homeostasis by regulating osmotic and oxidative substances to enhance tolerance to drought and salt stress in soybean [35]. The overexpression of *PtsrMYB* enhances tolerance to salt, dehydration, and cold in transgenic tobacco [36]. *MYB15*, *MYB37*, and *MYB96* were involved in the ABA-mediated inhibition of seed germination [37,38,39]. *GhODO1* positively regulates resistance to yellow wilt in cotton through the lignin biosynthesis and jasmonic acid signaling pathways [40]. These studies tentatively suggest that *PagMYB205* may be involved in the stress-related response of poplar.

To reveal the function of *PagMYB205*, we cloned *PagMYB205* from 84K poplar and cultivated transgenic poplar OX and RNAi. By subjecting transgenic plants to non-salt and salt stress treatment for 30d, it was found that there were mostly no significant differences between the morphology of transgenic and non-transgenic lines, but the rooting rate and root vitality of RNAi was higher than that of other lines under salt treatment conditions. These were consistent with other transcription factors in responding to abiotic stresses. For example, *LbTRY* was involved in root hair development and specifically reduced salt tolerance by increasing ion accumulation and reducing osmoregulatory substances when heterologously expressed in *Arabidopsis* [41]. The ectopic expression of *SiMYB75* promoted *Arabidopsis* root growth and regulated ABA-mediated drought and salt stress tolerance [42].

The overexpression or repression of transcription factors in plants affects the activity of related enzymes and changes in substance levels. *AcoMYB4* overexpression decreased POD, SOD, and CAT antioxidant enzyme activities, but significantly increased MDA levels [43]. The overexpression of *OsMYBR1* significantly increased soluble sugar and proline levels in transgenic rice [44], but *SiMYB56* decreased the MDA content [45]. The antioxidant system is essential for plant resistance by scavenging ROS to protect plants from oxidative damage and toxic effects [46]. POD, SOD, and CAT enzymes are all important protective enzymes in the antioxidant system [47]. SOD enzymes catalyze the disproportionation of ROS to non-toxic molecular oxygen, thus preventing plant toxicity, and POD and CAT enzymes remove hydrogen peroxide from the physiological system and also play a protective role [48,49]. The overexpression of *FvMYB82* resulted in higher proline and chlorophyll content, as well as higher SOD, POD, and CAT activities in *Arabidopsis* under salt and cold treatments, which improved salt and cold tolerance [50]. In this study, the POD, SOD, and CAT enzyme activities of RNAi under salt treatment were significantly higher than those of other lines, and those of RNAi3, with the lowest expression level of *PagMYB205*, were the highest. However, the enzyme activities of OX were lower or similar to those of WT. MDA is one of the membrane lipid peroxidation products and can be used as a sign to measure the degree of damage of cell membranes caused by abiotic stress [51]. It has been shown that the overexpression of *OsMYBR1* and *SiMYB56* improved the tolerance of rice to drought stress by increasing soluble sugar and proline content but decreasing MDA in transgenic rice plants [44,45]. In this study, the MDA content of RNAi was significantly lower than that of other lines under salt treatment conditions, and the lowest content was found in RNAi3. The above results indicated that the inhibiting expression of *PagMYB205* made a contribution to the salt stress tolerance of poplar; that is to say, PagMYB205 may negatively regulate the salt stress tolerance of poplar through interfering with ROS scavenging and root vitality (Figure 6).

## 4. Materials and Methods

### 4.1. Materials

The 84K poplars used in this study were grown on a 1/2 MS plant medium in a greenhouse (25 ± 2 °C, 60–70% relative humidity, 16/8 h light/dark cycle, with supplemental light of ~300 μEm^−2^s^−1^, three-band linear fluorescent lamp T5 28 W 6400 K). The roots, stems, and leaves used in this experiment were excised from the plants, immediately frozen in liquid nitrogen, and then stored at −80 °C [52].

### 4.2. Gene Cloning and Vector Construction

Total RNA was extracted using RNAprep pure Plant Kit from leaves of one-month-old plants (TIANGEN, Beijing, China). cDNA was synthesized using FastKing gDNA Dispelling RT SuperMix (TIANGEN, Beijing, China). *PagMYB205* was amplified using PF/R primers (Appendix A) and ligated into the pMD19-T vector (Takara, Beijing, China) via TA cloning for sequencing. *PagMYB205* without the stop codon was then amplified with the pBI121-PagMYB205-F/R primers (Appendix A) and inserted into the pBI121 vector via the *Xma I* and *Spe I* restriction sites to drive GFP expression. To construct the inhibited expression vector, the forward interfering fragment was amplified using primers CAM-RNAi-PagMYB205-F1/R1 (Appendix A) and ligated into the CAM-RNAi vector via the *Xho I* and *EcoR I* restriction sites. The reverse interfering fragment was then amplified using primers CAM-RNAi-PagMYB205-F2/R2 (Appendix A) and similarly ligated into the CAM-RNAi vector via the *Hind III* and *Xba I* restriction sites. Similarly, to analyze transcriptional activation activity, *PagMYB205* was amplified using primers PagMYB205-BD-F/R (Appendix A) and ligated into the pGBKT7 vector via the *EcoR I* and *Sal I* restriction sites. Three recombinant plasmids, 35S:PagMYB205-GFP, CAM-RNAi-PagMYB205, and pGBKT7-PagMYB205, were finally obtained.

### 4.3. Bioinformatic Analysis

The bioinformatics analysis was performed based on the sequences of CDs and amino acids of *PagMYB205*. ProtParam from the ExPASy database (https://www.expasy.org/resources/protparam, accessed on 16 February 2023) was used for the basic physicochemical properties analysis. The protein signal peptide, transmembrane structure, phosphorylation site, and secondary structure of proteins were analyzed with SignalP 5.0 (https://services.healthtech.dtu.dk/service.php?SignalP-5.0, accessed on 16 February 2023), TMHMM-2.0 (https://services.healthtech.dtu.dk/service.php?TMHMM-2.0, accessed on 16 February 2023), NetPhos 3.1 (https://services.healthtech.dtu.dk/service.php?NetPhos-3.1, accessed on 16 February 2023), and SOP 2021 (https://npsa-prabi.ibcp.fr/cgi-bin/npsa_automat.pl?page=npsa_sopma.html, accessed on 16 February 2023), respectively. The protein domain was identified with NCBI’s CD-search (https://www.ncbi.nlm.nih.gov/Structure/cdd/wrpsb.cgi, accessed on 16 February 2023). The 2000 bp upstream sequence of *PagMYB205* was obtained from PlantCare (http://bioinformatics.psb.ugent.be/webtools/plantcare/htmL/, accessed on 16 February 2023) and used for promoter cis-element analysis. The amino acid sequence of homologous genes of *PagMYB205* was blasted and downloaded from NCBI (https://www.ncbi.nlm.nih.gov/, accessed on 16 February 2023) to construct a neighbor-joining phylogenetic tree and amino acid sequence comparison using BioEdit 7.0.9.1 [53].

### 4.4. Subcellular Localization

*Agrobacterium* containing subcellular localization vectors 35S:PagMYB205-GFP or no-load control 35S:GFP were injected into tobacco leaves using 5 mL sterile needleless syringes. After being grown in the dark for 2–3 day, the fluorescence signals under different fields of view were observed and photographed under a confocal laser scanning microscope (Olympus FV1000, Olympus, Tokyo, Japan).

### 4.5. Transactivation Activity Assay

To verify whether PagMYB205 has self-activating activity, we fused the *PagMYB205* gene into the pGBKT7 vector to obtain the pGBKT7-PagMYB205 plasmid. Yeast cells transfected with the pGBKT7 empty vector and the pGBKT7-53/pGBKT7-T were designed as the negative and positive control, respectively. Yeast cells were incubated on SD/-Trp and SD/-Trp/-His/-Ade/X-α-Gal solid media to observe the yeast growth status and to test the self-activation ability.

### 4.6. Generation and Identification of Transgenic Poplar

Transgenic poplars were obtained by using the *Agrobacterium*-mediated transformation method [54]. The transgenic plants were verified in terms of DNA and RNA levels using PCR and RT-qPCR. Primers PagMYB205-F/R (Appendix A) were used to determine the relative expression of *PagMYB205* in each line using SuperReal PreMix Plus (SYBR Green, TIANGEN, Beijing, China) with *actin* as the internal reference gene. The fold change of gene expression was calculated by using the 2^−ΔΔCT^ method.

### 4.7. Salt Tolerance Assay

Salt tolerance tests were performed using transgenic poplar and WT. For each line, 30-day-old plants of uniform height and vigorous growth were selected, and the terminal buds were cut and subcultured in 1/2 MS medium containing 0 or 50 mmol/L NaCl. After 30 d, fresh weight, plant height, and number of leaves were statistically determined. The roots were scanned and photographed using an EPSON EXPRESSION 10,000 XL root scanning system (resolution set at 400 dpi) to detect the primary root length, root diameter, root area, root volume, etc., and WinRHIZO 2016p software (Regent Instruments Inc., Quebec, QC, Canada) was used for statistical analysis [55]. The 3rd function leaf from top to bottom was isolated, and the stomata on the back of the leaf were fixed with clear nail polish and observed using microscopy (Leica DM6B, Leica, Wetzlar, Germany). At least six biological replicates were performed for each experiment.

### 4.8. Determination of Physiological Indicators

Leaves of 30 day seedlings were selected for the determination of relevant physiological indices. SOD activity was measured using the nitrogen blue tetrazolium (NBT) photoreduction method. CAT activity was determined using UV spectrophotometry and POD activity was determined using the guaiacol method. MDA content was determined using the colorimetric method using barbituric acid. Root vitality was determined using the triphenyltetrazolium chloride (TTC) method [56,57]. At least six biological replicates were performed for each experiment.

### 4.9. Statistical Analysis

The data were processed using SPSS 22.0 software (Chicago, IL, USA). *p* < 0.05 was used to test the significance of differences between the experimental and control groups. Plots were made using Origin 2019b (OriginLab, Northampton, MA, USA) and Adobe Photoshop 2022. Each experiment was performed with three technical repeats.

## 5. Conclusions

PagMYB205 protein was an unstable hydrophilic soluble protein, localized in the nucleus without a transmembrane structural domain and self-activating activity. PagMYB205 had cis-acting elements related to abiotic stresses in the promoter region. Inhibiting the expression of *PagMYB205* reduced the MDA content of the RNAi, but increased SOD and CAT enzyme activities and root vitality under salt stress. Above all, this study preliminarily revealed the negative function of PagMYB205 in poplar salt stress responses.

## Figures and Tables

**Figure 1 ijms-24-15437-f001:**
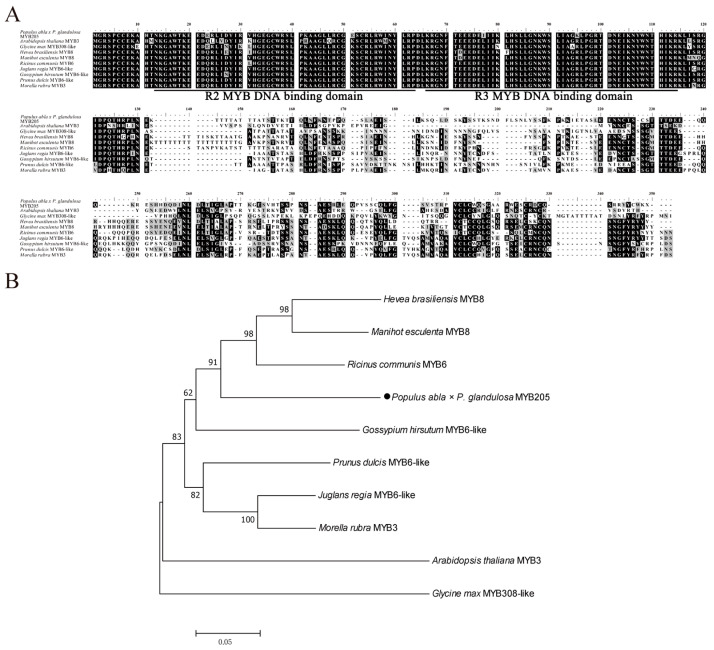
Multiple sequence alignment and phylogenetic analysis of PagMYB205 protein. (**A**) Multiple alignment of PagMYB205 amino acid sequences was performed with BioEdit. Black indicates identical amino acids and grey indicates similar amino acids. (**B**) Phylogenetic trees of PagMYB205 were constructed using Clustalx 1.83 and MEGA 7.0 with the neighbor-joining methods.

**Figure 2 ijms-24-15437-f002:**
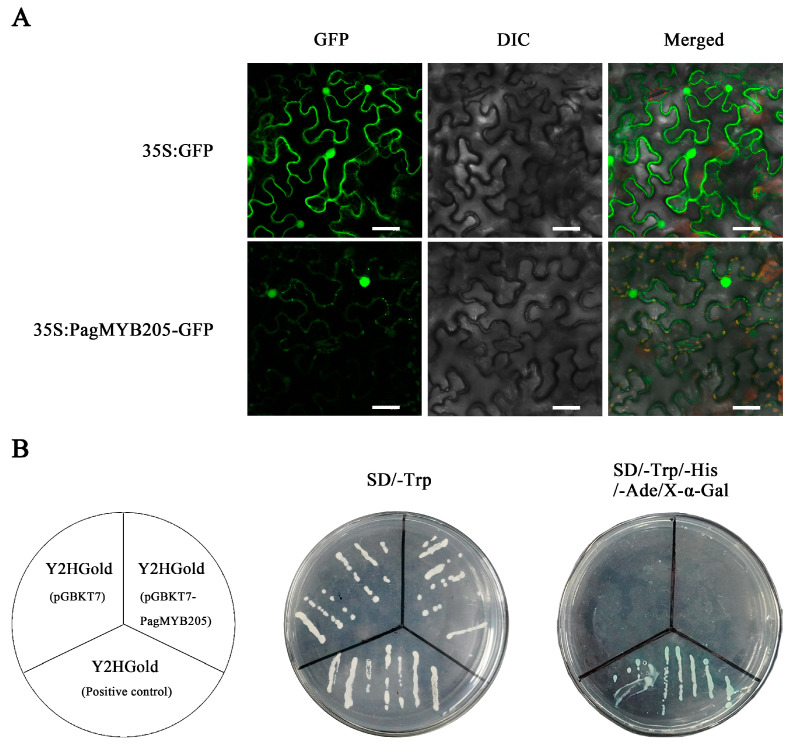
Subcellular localization and transactivation assay of PagMYB205. (**A**) Subcellular localization of the PagMYB205 protein. The recombinant vector (35S:PagMYB205-GFP) and the control vector (35S:GFP) were transiently expressed in tobacco leaves. Scale bar = 40 μm. GFP: green fluorescent protein; DIC: differential interference contrast. (**B**) Transactivation assay of PagMYB205 in yeast cells.

**Figure 3 ijms-24-15437-f003:**
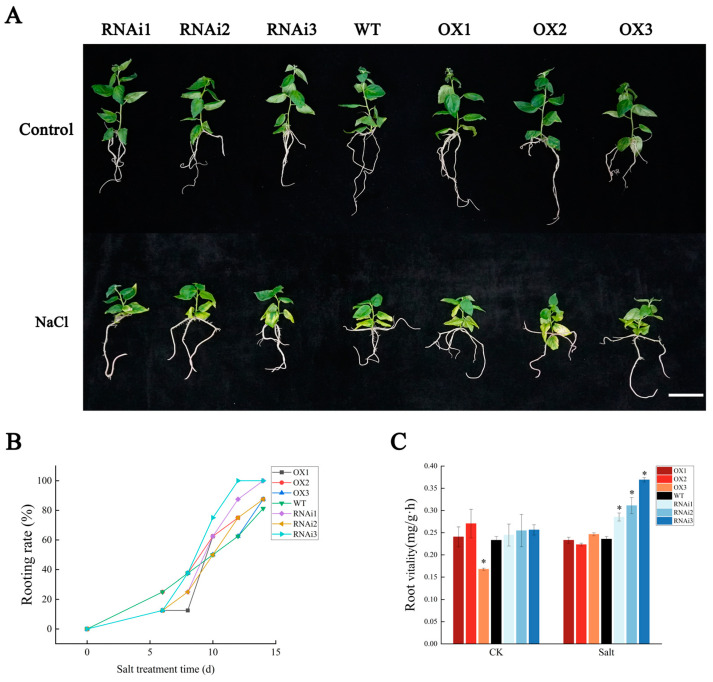
Morphological characteristics of transgenic poplars under salt stress. (**A**) Phenotypes of one-month-old poplars cultured in 1/2 MS medium containing 0 and 50 mmol/L NaCl, respectively. Scale bar = 5 cm. (**B**) Rooting rate of transgenic plants at different time points under salt stress. (**C**) Root vitality of transgenic plants after salt stress treatment. The data represent the mean ± SE from six biological replicates; * indicates significant difference between transgenic line and WT (*p* < 0.05). OX: *PagMYB205* overexpression lines; WT: wild-type (non-transgenic) poplar; RNAi: *PagMYB205* inhibited expression lines; CK: normal conditions; Salt: salt stress conditions.

**Figure 4 ijms-24-15437-f004:**
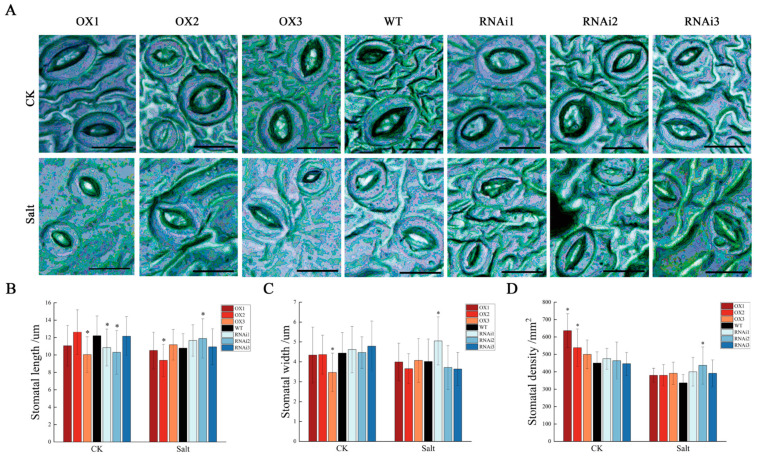
Analysis of stomatal morphological indices. (**A**) Stomatal morphological characteristics under salt stress. (**B**–**D**) Statistic analysis of stomatal length, stomatal width, and stomatal density between different lines. Scale bar = 20 μm. The data represent the mean ± SE from six biological replicates; * indicates significant difference between transgenic line and WT (*p* < 0.05). OX: *PagMYB205* overexpression lines; WT: wild-type (non-transgenic) poplar; RNAi: *PagMYB205* inhibited expression lines; CK: normal conditions; Salt: salt treatment conditions.

**Figure 5 ijms-24-15437-f005:**
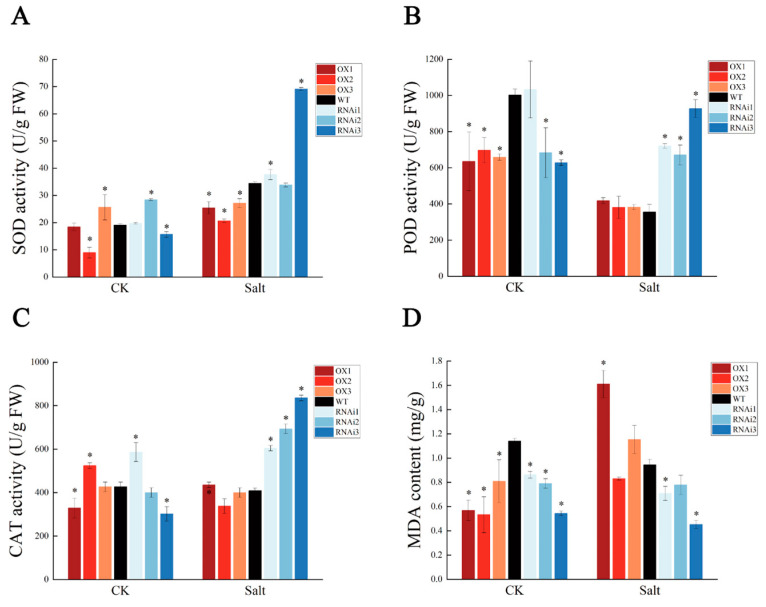
PagMYB205 interferes with ROS scavenging of poplar. (**A**–**C**) SOD, POD, and CAT activities of transgenic lines under salt stress. (**D**) MDA content of transgenic lines under salt stress. The data represent the mean ± SE from six biological replicates; * indicates significant difference between transgenic line and WT (*p* < 0.05). OX: *PagMYB205* overexpression lines; WT: wild-type (non-transgenic) poplar; RNAi: *PagMYB205* inhibited expression lines; CK: normal conditions; Salt: salt treatment conditions.

**Figure 6 ijms-24-15437-f006:**
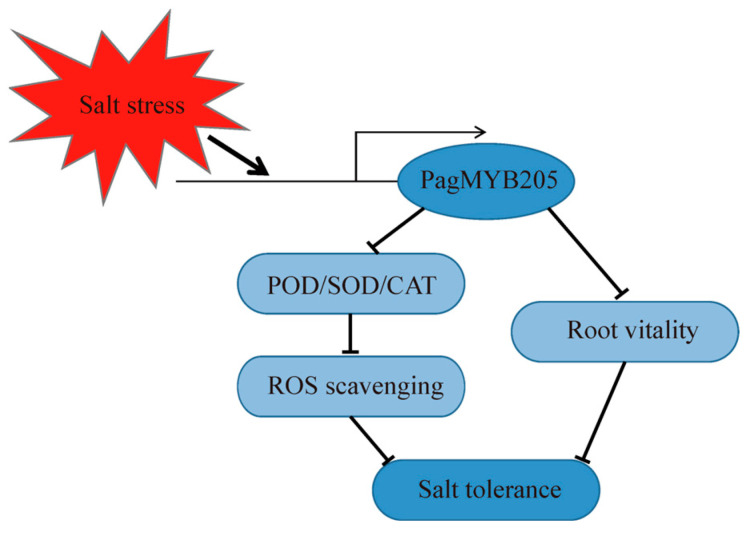
The model of PagMYB205 negatively regulated the salt tolerance ability of poplar. PagMYB205 was up-regulated by salt stress induction. Increased expression of PagMYB205 interfered with the ability of ROS scavenging, as well as root vitality, to negatively regulate the salt tolerance of poplar.

## Data Availability

All of the data generated or analyzed during this study are included in this published article and its Appendix A.

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
