# Peer review of "PagMYB205 Negatively Affects Poplar Salt Tolerance through Reactive Oxygen Species Scavenging and Root Vitality Modulation"

_ijms, 2023, doi:10.3390/ijms242015437_

Round 1

Reviewer 1 Report

The present document evaluates the impact of PagMYB205 on poplar salt tolerance modifing the roots behavior.

The introduction, except the first two references (should be changed), which are not really convenient and adapted to the sentences they are supporting, the rest is fine, complete and present clearly the problem.

The material and method section is also complete and clear.

The results and discussion are clearly presented, only some quality of figures should be improved; The discussion goes in detail on the analysis of the results.

The conclusion is in line with the results obtained. The abstract is good.

Minor comments can be found in the attached document

Author Response

Dear Editors and Reviewers:

Thank you for your comments concerning our manuscript entitled “PagMYB205 negatively affects poplar salt tolerance through ROS scavenging and root vitality modulation” (ID: ijms-2667191). Those comments are all valuable and very helpful for revising and improving our paper. We have studied comments carefully and have made correction which we hope meet with approval. Revised portions are marked in the tracked changes manuscript. The main corrections in the paper and the responds to the comments are as follows:

Responds to the reviewer’s comments:

Reviewer 1:

The present document evaluates the impact of PagMYB205 on poplar salt tolerance modifing the roots behavior.

The introduction, except the first two references (should be changed), which are not really convenient and adapted to the sentences they are supporting, the rest is fine, complete and present clearly the problem.

Response: Thank reviewer for pointing this out. According to the reviewer's advice, we have changed the first two references.

Soil salinisation is one of the main environmental stresses limiting plant growth and production [1]. Research has shown that 20% of the world's arable land and nearly half of the irrigated land are affected by salt stress in varying degrees [2].

  1. Zhang, Y.; Xu, J.; Li, R.; Ge, Y.; Li, Y.; Li, R. Plants’ Response to Abiotic Stress: Mechanisms and Strategies. 2023, 24, 10915, doi:10.3390/ijms241310915.
  2. Hasanuzzaman, M.; Raihan, M.R.H.; Masud, A.A.C.; Rahman, K.; Nowroz, F.; Rahman, M.; Nahar, K.; Fujita, M. Regulation of Reactive Oxygen Species and Antioxidant Defense in Plants under Salinity. 2021, 22, 9326, doi:10.3390/ijms22179326.

-----Line 33-34, 383-387.

The material and method section is also complete and clear.

The results and discussion are clearly presented, only some quality of figures should be improved; The discussion goes in detail on the analysis of the results.

The conclusion is in line with the results obtained. The abstract is good.

Response: Thank reviewer for the positive comment. According to the reviewer's advice, we have uploaded a clearer figure.

Minor comments:

(1) useless if already cited in the title, latin name?

Response: Thank reviewer for pointing this out. According to the reviewer's advice, we have changed it.

-----Line 28.

(2) This reference is not the best one adapted to the sentence you want to support, as those are not the results of this publication

Response: Thank reviewer for pointing this out. We have changed the first two references.

-----Line 33-34, 383-387.

(3) high salt concentration?

Response: Thank reviewer for pointing this out. We have added the word.

-----Line 42.

(4) compared to?

Response: Thank reviewer for pointing this out. We have changed the word.

-----Line 44.

(5) high salt level? concentration?

Response: Thank reviewer for pointing this out. We have added the word.

-----Line 50.

(6) replace by a comma?

Response: Thank reviewer for pointing this out. We have replaced it.

-----Line 53.

(7) The quality of the image is not enough good, while we do a zoom we have difficulties to read correctly

Response: Thank reviewer for pointing this out. According to the reviewer's advice, we have uploaded a clearer figure.

(8) please indicate in the legend what this means

Response: We have indicated this in the legend.

-----Line 131-132.

We tried our best to improve the manuscript and made some changes in the manuscript. These changes will not influence the content and framework of the paper.

We appreciate for Editors/Reviewers’ warm work earnestly, and hope that the correction will meet with approval.

Once again, thank you very much for your comments and suggestions.

Yours sincerely,

Shengji Wang

Reviewer 2 Report

Review for

 PagMYB205 negatively affects poplar salt tolerance through ROS scavenging and root vitality modulation

by Lieding Zhou 1, Xuhui Huan 1, Kai Zhao 1, Xia Jin 1, Jia Hu 1, Shuhui Du 1, Youzhi Han 1, Shengji Wang 1,* 4

 Salt stress is one of the major abiotic stresses that limits plant growth and development.  The MYB transcription factor family plays essential roles in plant growth and development, as well  as stress tolerance process.

resistance of poplar to salt is an active research area

with different molecular targets

PabCIPK24 plays an important role in the response of hybrid poplar to salt stress

Bai, X.-D.Wang, W.Ji, J.-B., ...Yang, C.-P.Liu, G.-F.

Industrial Crops and Products, 2023, 205, 117452

PagMYB151 facilitates proline accumulation to enhance salt tolerance of poplar

Hu, J.Zou, S.Huang, J., ...Han, Y.Wang, S.

BMC Genomics, 2023, 24(1), 345

PagDA1a and PagDA1b expression improves salt and drought resistance in transgenic poplar through regulating ion homeostasis and reactive oxygen species scavenging

Zhao, Y.Lu, K.Zhang, W., ...Yang, Q.Zhang, H.

Plant Physiology and Biochemistry, 2023, 201, 107898

PeCLH2 Gene Positively Regulate Salt Tolerance in Transgenic Populus alba × Populus glandulosa

Ge, X.Du, J.Zhang, L.Qu, G.Hu, J.

Genes, 2023, 14(3), 538

--------------

 transgenic poplar  lines with PagMYB205 overexpression (OX) or inhibited expression (RNAi, RNA interference) were  cultivated

could you please update the readers about the current regulatory status for large scale cultivation of transgenic poplar  lines

in many countries worldwide

--------------------

Poplar is an important raw material for biofuels and widely used as green forest,  timber forest and protection forest for production, road greening and desertification control [24].

fully true

--------------

With the increase of soil salinization, salt stress has become an important environmental factor limiting the growth of poplar [25,26].

please add a few words about the sources, the events responsible for the increase of soil salinization

----------------

Therefore, there is an urgent need to breed new poplar species of salt-tolerance.

or urgent need to reduce soil salinization

---------------

very nice study

discussion supported by accurate experimental data

Author Response

Dear Editors and Reviewers:

Thank you for your comments concerning our manuscript entitled “PagMYB205 negatively affects poplar salt tolerance through ROS scavenging and root vitality modulation” (ID: ijms-2667191). Those comments are all valuable and very helpful for revising and improving our paper. We have studied comments carefully and have made correction which we hope meet with approval. Revised portions are marked in the tracked changes manuscript. The main corrections in the paper and the responds to the comments are as follows:

Responds to the reviewer’s comments:

Reviewer 2:

transgenic poplar lines with PagMYB205 overexpression (OX) or inhibited expression (RNAi, RNA interference) were cultivated

could you please update the readers about the current regulatory status for large scale cultivation of transgenic poplar lines

in many countries worldwide

Response: Thank reviewer for pointing this out. We have added instructions.

Currently, genetically modified poplar is still in field trials in the United States and Europe due to the rigorous environmental risk assessments required for regulatory approval. China is the only country in the world that has approved commercial cultivation of genetically modified poplar [25,26].

  1. Porth, I.; El-Kassaby, Y. Current status of the development of genetically modified (GM) forest trees world-wide: a com-parison with the development of other GM plants in agriculture; CABI: 2014; Volume 2014, pp. 1–12, doi:10.1079/PAVSNNR20149008.
  2. Clifton-Brown, J.; Harfouche, A.; Casler, M.D.; Dylan Jones, H.; Macalpine, W.J.; Murphy-Bokern, D.; Smart, L.B.; Adler, A.; Ashman, C.; Awty-Carroll, D.; et al. Breeding progress and preparedness for mass-scale deployment of perennial lignocellulosic biomass crops switchgrass, miscanthus, willow and poplar. 2019, 11, 118-151, doi:10.1111/gcbb.12566.

-----Line 85-88, 443-448.

Poplar is an important raw material for biofuels and widely used as green forest, timber forest and protection forest for production, road greening and desertification control [24].

fully true

Response: Thank reviewer for the positive comment.

With the increase of soil salinization, salt stress has become an important environmental factor limiting the growth of poplar [25,26].

please add a few words about the sources, the events responsible for the increase of soil salinization

Response: Thank reviewer for pointing this out. We have added instructions.

As a result of land clearing, unsustainable irrigation practices, and pressure to bring marginal land into production, salinity stress has become an important environmental factor limiting poplar growth [27,28].

-----Line 88-90.

We tried our best to improve the manuscript and made some changes in the manuscript. These changes will not influence the content and framework of the paper.

We appreciate for Editors/Reviewers’ warm work earnestly, and hope that the correction will meet with approval.

Once again, thank you very much for your comments and suggestions.

Yours sincerely,

Shengji Wang